# Adults with depressive symptoms have lower odds of dietary supplement use

**Shakila Meshkat**[1], **Vanessa K. Tassone**[1], **Hilary Pang**[2], **Michelle Wu**[1], **Hyejung Jung**[3], **Wendy Lou**[3], **Venkat Bhat**[1,2,4,5]*

1 Interventional Psychiatry Program, St. Michael's Hospital, Toronto, Ontario, Canada, 2 Department of Psychiatry, University of Toronto, Toronto, Ontario, Canada, 3 Department of Biostatistics, Dalla Lana School of Public Health, University of Toronto, Toronto, Ontario, Canada, 4 Institute of Medical Science, Temerty Faculty of Medicine, University of Toronto, Toronto, Ontario, Canada, 5 Mental Health and Addictions Services, St. Michael's Hospital, Toronto, Ontario, Canada

* venkat.bhat@utoronto.ca

**Data Availability Statement:** The data underlying the results presented in the study are available from https://wwwn.cdc.gov/nchs/nhanes/Default.aspx. We used data from 2005-2006 through 2017-2018 survey cycle datasets.

## Abstract

### Objective

In this study, we aim to evaluate dietary supplement and complementary and alternative medicine (CAM) use in individuals with depressive symptoms. Furthermore, we conducted a comparative analysis of the usage of these agents among individuals with depressive symptoms, differentiating between those who were using antidepressants and those who were not. Additionally, we compared individuals with depressive symptoms who were not using antidepressants with participants who did not have depressive symptoms as well as individuals with depressive symptoms who were using antidepressants with individuals without depressive symptoms.

### Method

The National Health and Nutrition Examination Survey 2007–2018 data was collected. Depressive symptoms were assessed using patient health questionnaire-9. Dietary supplement and antidepressants use was evaluated using Dietary Supplement Use and Prescription Medications Questionnaires.

### Results

31,445 participants, with 2870 (8.05%) having depressive symptoms were included. Participants with depressive symptoms had significantly lower odds of dietary supplement use compared with those without depressive symptoms (aOR = 0.827, 95% CI: 0.700,0.977, p = 0.026). Participants with depressive symptoms who were using antidepressants had significantly higher odds of dietary supplement (aOR = 1.290, 95% CI: 1.038,1.604, p = 0.022) compared with participants with depressive symptoms who were not using antidepressants. Furthermore, Participants with depressive symptoms who weren't using antidepressants had significantly lower odds of dietary supplement use (aOR = 0.762, 95% CI: 0.632,0.918, p = 0.005) compared with participants without depressive symptoms. In individuals with

**Funding:** The authors received no specific funding for this work.

**Competing interests:** I have read the journal's policy and the authors of this manuscript have the following competing interests: VB is supported by an Academic Scholar Award from the University of Toronto Department of Psychiatry and has received research support from the Canadian Institutes of Health Research, Brain & Behavior Foundation, Ontario Ministry of Health Innovation Funds, Royal College of Physicians and Surgeons of Canada, Department of National Defence (Government of Canada), New Frontiers in Research Fund, Associated Medical Services Inc. Healthcare, American Foundation for Suicide Prevention, Roche Canada, Novartis, and Eisai. This does not alter our adherence to PLOS ONE policies on sharing data and materials.

treated depressive symptoms compared to those without depressive symptoms, CAM use was significantly lower (aOR = 0.763, 95% CI = 0.598,0.973, p = 0.030).

## Conclusion

Individuals with depressive symptoms have lower odds of dietary supplement use. Further studies are needed to replicate these findings and examine the underlying mechanisms for this association.

## 1. Introduction

Major depressive disorder (MDD) is among the leading causes of disability with a prevalence of 280 million globally [1]. MDD is not a single, homogeneous disorder but a complex phenomenon with many subtypes and probably several etiologies [2]. It also involves connections with other psychiatric and somatic disorders, a propensity for episodic and frequently progressive mood disturbances, a range of symptomatology from moderate to severe symptoms with or without psychotic aspects, and variances in symptomatology [3]. Depression is associated with increased morbidity and mortality rates with treatment involving adherence to psychotherapy and/or medication [1, 2]. Numerous pathways, such as synaptic transmission, the monoamine hypothesis, neurotransmitter concentrations, transporters for neurotransmitter reuptake, and neurotransmitter receptors have been identified as being involved in the pathophysiology of depression [4]. Additionally, nutrition may play a significant role in depression prevention, with involvement with the gut microbiome [5]. Nutritional deficiencies can have an impact on brain functioning, leading to mood disorders such as depression, dysthymia, and anxiety disorders [6]. Additionally, depression has been linked to deficits in several vitamins such as vitamin D, folic acid, vitamin B12, niacin, and vitamin C [7].

Dietary supplements are chemical substances such as minerals, vitamins, and anti- oxidants, which are part of normal nutrition but also can be added to normal nutrition in the shape of more or less pure substances [8]. Despite the fact that dietary supplements are not recommended for individuals who are healthy and consume a well-balanced and nutritionally adequate diet, the industry generates an annual revenue of 1.2 billion USD [9]. Additionally, it is estimated that approximately 29.2% of people in the United States (US) use at least one dietary supplement [8]. Dietary supplement use is more common in women and people with healthy lifestyles and reduces risk of chronic medical conditions [10, 11]. As more is learned about the therapeutic effects of dietary components, dietary supplements are increasingly being developed for the treatment of specific medical conditions including mental disorders. However, there are limited studies on examining the utilization of dietary supplements in people with depression at the population-level.

Complementary and alternative medicine (CAM) refers to a diverse range of medical and healthcare systems, practices, and products that are currently not included in conventional medicine [12]. Depression is one of the 10 most frequent indications for using CAM since it is often undertreated and under diagnosed [13]. Despite the advantages associated with CAM, such as a lower incidence of adverse events, reported efficacy, a comprehensive perspective on individual problems, and dissatisfaction with traditional care, the quality of CAM recommendations for depression has been noted to be generally lower [14]. Herbal agents, exercise, and dietary supplements are mostly used as CAM by depressed individuals [15]. While different

agents are used in depression as CAM, there is limited evidence regarding the use of each agent in individuals with depression.

Given the widespread use of CAM and dietary supplements, we aim to investigate 1) the association between use of these supplement agents and depressive symptom severity, 2) if individuals with depressive symptoms use these agents more than individuals without depressive symptoms, 3) if individuals with depressive symptoms who use antidepressants use these agents more than individuals with depressive symptoms who do not use antidepressants, 4) if individuals with depressive symptoms who do not use antidepressants use these agents more than individuals without depressive symptoms, and 5) if individuals with depressive symptoms who use antidepressants use these agents more than individuals without depressive symptoms.

## 2. Method

### 2.1. Study population and design

We used data from the National Health and Nutrition Examination Surveys (NHANES) 2007–2018. NHANES contains cross sectional population based data that was conducted by the US National Center for Health Statistics (NCHS), Center for Disease Control and Prevention (CDC). NHANES is a survey based nationally data from civilian non-institutionalized populations aged 18–80 years in 50 states of the United States that has complex, multistage, probability sampling design. The data is collected through in-home interviews using computer-assisted personal interviews conducted at home and physical examinations performed at mobile examination centers (MEC). Socio-demographic, nutritional, and health-related inquiries were addressed during in-home interviews. Anthropometric, blood pressure, urine, blood, and hearing tests, as well as health status evaluations, were done at the MEC. Before conducting in-home interviews and the MEC examinations, participants would need to sign consent forms [16]. The survey procedure has been approved by the NCHS Research Ethics Review Board. NHANES data is collected annually and released every two years. Additional details about methods and sampling are presented on the CDC website: https://wwwn.cdc.gov/nchs/nhanes/. We included male and female older than 18 years old who completed the Mental Health—Depression Screener (items DPQ010 to DPQ090), Dietary Supplement Use questionnaire and Prescription Medications—Drug Information Questionnaire.

### 2.2. Exposure variables

Depressive symptoms were assessed using the Patient Health Questionnaire-9 (PHQ-9). PHQ-9 is a self-administered questionnaire widely used for making criteria-based diagnoses of depressive symptoms in accordance with the *Diagnostic and Statistical Manual of Mental Disorders*, *Fourth Edition* for MDD [17]. On a scale of 0 to 3, participants rated each question as per the frequency of their symptoms during the previous two weeks, with 0 being not at all and 3 being experienced nearly every day. While a score of less than 10 shows no depressive symptoms, a score of 10 or higher suggests presence of depressive symptoms [18]. Several studies have confirmed the reliability and validity of PHQ-9 [17, 19].

Antidepressants use was evaluated using Prescription Medications—Drug Information Questionnaire. Prescription medications were classified by NHANES based on the three-level nested therapeutic classification scheme from Cerner Multum's Lexicon (https://www.cerner.com/solutions/drug-database). Antidepressants were selected by a second-level category identifier of "249" (https://wwwn.cdc.gov/Nchs/Nhanes/1999-2000/RXQ_DRUG.htm). Binary variables were then constructed for depressive symptoms treatment, where "treated" indicates anyone with depressive symptoms and currently on any class of antidepressants and "untreated" for those diagnosed with depressive symptoms but not taking any class of

antidepressants. Depressive symptoms severity was defined as just the raw PHQ-9 score, where a higher score indicates more depressive symptoms.

## 2.3. Outcome variable

Dietary supplements use during the past 30 days was self reported, including vitamin B1, B2, B3 (Niacin), B6, B12, C, K, D, folic acid, folate, choline, calcium, phosphorus, magnesium, iron, zinc, copper, sodium, potassium, selenium, vitamin E, omega-3 fatty acids, and multivitamins. Dietary supplement use was binary and was considered "Yes" if individuals were taking any of the previously listed supplements as well as answered "Yes" to DSD010 "Any dietary supplements taken?", and "No" otherwise. Selected CAM agents were: folate, magnesium, vitamin B6, B9, D, zinc, selenium, and omega-3 fatty acids. In this study, CAM was defined as a binary variable of "Yes" if any of the above CAM agents were taken or if an individual participated in >150 minutes of physical activity a week, and "No" otherwise. We also categorized supplements into vitamin B family (vitamin B1, B2, niacin, folate, B12, folic acid), fat-soluble vitamins (D, E, K) and water-soluble vitamins (vitamin C, B1, B2, B3, folate, folic acid). These were all binary variables that were dichotomized into "Yes" or "No" for specific vitamin-class use, with "No" set as the reference category. Individual vitamin use was kept continuous in their respective units. Note that while the majority of the vitamin and supplements information comes from the Dietary Supplement Use 30-Day—Total Dietary Supplements Questionnaire, information on vitamin E, omega 3 fatty-acids, and multivitamin use had to be extracted from the Prescription Medications questionnaire and thus details on exact amounts taken per individual was not available.

## 2.4. Covariates

In order to minimize confounding bias, covariates were included. Covariates included age, race, sex, diabetes, hypertension, chronic kidney disease, congestive heart failure, and liver disease. Age was continuous and gender was binary (male, female). All other health indicator variables were binary where "Yes" indicates presence of the disease and "No" otherwise. In models where depressive symptoms are treated as a binary variable, depressive symptoms severity (total PHQ-9 score) was included as a control.

## 2.5. Statistical analysis

All statistical analyses were performed using R v 4.2.2 and the package "survey" to account for MEC survey weights (https://wwwn.cdc.gov/nchs/nhanes/tutorials/weighting.aspx). Survey weights were divided by the number of cycles included to account for the merging of cycles, as recommended by NHANES. Categorical variables were described as raw frequency and weighted percent in the study population demographic characteristics table, while continuous variables were described as weighted mean and standard deviation (*SD*). A chi-square test of independence was used to check for statistically significant ($p$-value $\leq 0.05$) differences in categorical demographic characteristics among the depressive symptoms groups, and a t-test was used to test for differences in continuous variables. Multivariable logistic regression was used to test the relationship between dietary supplement and CAM use with depressive symptoms, depressive symptoms severity, treated vs. untreated depressive symptoms, and untreated depressive symptoms vs. no depressive symptoms. The same methodology was applied with the use of vitamin classes as the outcome variable. Multiple linear regression was used to look at the association between individual vitamins and depressive symptoms, depressive symptoms severity, treated vs. untreated depressive symptoms, and untreated vs. no depressive symptoms. To account for multiple testing, statistical significance was adjusted using Bonferroni's

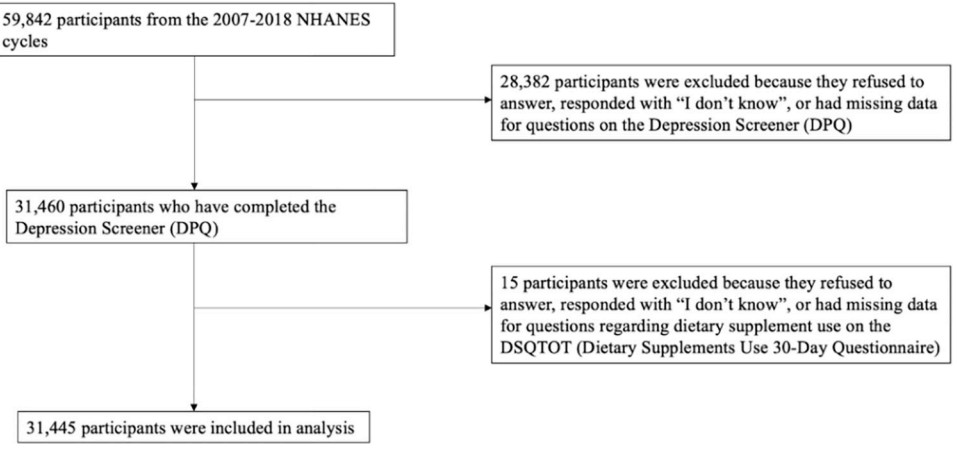

**Fig 1. Participant inclusion flowchart.**

correction for the individual vitamins models and the vitamin class models. Missing values were excluded from the analyses.

## 3. Results

### 3.1. Study sample

Data from a total of 59,842 individuals aged 18 to 80 years were collected from the 2007–2018 NHANES cycles. The study population consisted of 31,445 individuals (Fig 1), of which 2870 (8.05%) reported depressive symptoms, 15518 (54.32%) reported using dietary supplements, and 22332 used CAM (76.42%). The mean age of the study population was 46.66 (SD = 17.50), and 15958 (51.13%) were females. Table 1 presents the demographic characteristics for the study population and any significant differences among covariates.

### 3.2. Dietary supplement use

In adjusted models, participants with depressive symptoms had significantly lower odds of dietary supplement use in comparison with those without depressive symptoms. The adjusted model showed that those with depressive symptoms have 17.3% lower odds of using dietary supplement compared to those without depressive symptoms (adjusted odds ratio [aOR] = 0.827, 95% CI: 0.700, 0.977, p = 0.026). However, depressive symptoms severity did not show this pattern (Table 2). Conversely, between treated and untreated individuals with depressive symptoms, those who were treated had statistically significant higher odds of using dietary supplements compared to those who were not treated (aOR = 1.290, 95% CI: 1.038,1.604, p = 0.022) (Table 3). For patients with untreated depressive symptoms, odds of dietary supplement use was significantly lower compared to those without depressive symptoms (aOR = 0.762, 95% CI: 0.632,0.918, p = 0.005) (Table 3). No statistically significant association was seen between those with treated depressive symptoms versus no depressive symptoms in terms of dietary supplement usage (Table 3).

### 3.3. Complementary and alternative medicine

Participants with depressive symptoms had lower odds of CAM use in comparison with those without depressive symptoms (aOR = 0.831, 95% CI: 0.710, 0.977, p = 0.026). Moreover, participants with more severe depressive symptoms had significantly lower odds of CAM use in

**Table 1. Demographics of study sample.**

| Characteristic | Total | Depressive symptoms | No-depressive symptoms | P value |
|---|---|---|---|---|
| Sample size | 31445 | 2870 | 28575 | |
| Age (mean, *SD*) | 46.66 (17.50) | 46.13 (16.43) | 46.71 (17.59) | 0.227 |
| Sex (n, %) | | | | <0.001 |
| Female | 15958 (51.13) | 1827 (64.16) | 14131 (49.98) | |
| Male | 15487 (48.87) | 1043 (35.84) | 14444 (50.02) | |
| Race (n, %) | | | | <0.001 |
| Mexican American | 4857 (8.71) | 440 (8.29) | 4417 (8.74) | |
| Other Hispanic | 3280 (5.82) | 397 (8.09) | 2883 (5.62) | |
| Non-Hispanic White | 12864 (66.66) | 1190 (62.94) | 11674 (66.98) | |
| Non-Hispanic Black | 6786 (11.16) | 613 (13.20) | 6173 (10.98) | |
| Other race—including multi-racial | 3658 (7.66) | 230 (7.48) | 3428 (7.68) | |
| Diabetes—Yes (n, %) | 4042 (9.69) | 570 (15.61) | 3472 (9.17) | <0.001 |
| Hypertension—Yes (n, %) | 8903 (25.56) | 1148 (37.26) | 7755 (24.53) | <0.001 |
| Chronic Kidney Disease—Yes (n, %) | 1016 (2.65) | 207 (6.60) | 809 (2.30) | <0.001 |
| Congestive Heart Failure—Yes (n, %) | 965 (2.37) | 189 (5.69) | 776 (2.08) | <0.001 |
| Liver Disease—Yes (n, %) | 1257 (3.69) | 255 (8.36) | 1002 (3.28) | <0.001 |
| Dietary supplement use (n, %) | 15518 (54.32) | 1368 (53.40) | 14150 (54.40) | 0.470 |
| CAM use (n, %) | 22332 (76.42) | 1762 (66.34) | 20570 (77.30) | 0.062 |
| Vitamin B family (n, %) | 10230 (36.28) | 810 (31.11) | 9420 (36.73) | <0.001 |
| Water-soluble vitamins (n, %) | 10777 (38.19) | 831 (31.93) | 9946 (38.74) | <0.001 |
| Fat-soluble vitamins (n, %) | 11054 (39.24) | 881 (33.43) | 10173 (39.75) | <0.001 |
| Treated depressive symptoms | 910 (36.53) | 910 (36.53) | N/A | N/A |

Note: All percentages, means and SDs are using survey weights. *P* values are from t-tests for weighted continuous variables and chi-squared tests for weighted categorical variables.

comparison with those milder depressive symptoms (aOR = 0.966, 95% CI: 0.957, 0.974, p = <0.001) (Table 2). Between treated and untreated individuals with depressive symptoms though, odds of CAM use was not statistically significantly different than those with treated depressive symptoms compared to those with untreated depressive symptoms (Table 3). Results were also not significant when comparing those with untreated depressive symptoms to those without depressive symptoms (Table 3). In those with treated depressive symptoms compared to no depressive symptoms, CAM use was lower in comparison (aOR = 0.763, 95% CI = 0.598,0.973, p = 0.030) (Table 3).

**Table 2. Unadjusted and adjusted ORs for associations between dietary supplement and complementary and alternative medicine use and depressive symptoms.**

| Exposure group | OR (95% CI) | *P* value | aOR (95% CI) | *P* value |
|---|---|---|---|---|
| **Dietary supplement** | | | | |
| Depressive symptoms vs. No depressive symptoms | 0.960 (0.859,1.074) | 0.475 | 0.8272 (0.700,0.977) | 0.026* |
| Depressive symptoms severity | 1.001 (0.994,1.009) | 0.727 | 0.998(0.990,1.007) | 0.702 |
| **Complementary and alternative medicine** | | | | |
| Depressive symptoms vs. No depressive symptoms | 0.579 (0.521,0.642) | <0.001* | 0.831(0.710,0.974) | 0.023* |
| Depressive symptoms severity | 0.960 (0.952,0.968) | <0.001* | 0.966(0.957,0.974) | <0.001* |

Note. aOR = adjusted odds ratio. Covariates in the adjusted model include age, gender, diabetes, hypertension, chronic kidney disease, congestive heart failure, and liver disease. Statistical significance of $p < 0.05$ is indicated by *.

**Table 3. Unadjusted and adjusted ORs for associations between dietary supplement and complementary and alternative medicine use and depressive symptoms treatments.**

| Outcome | N | OR (95% CI) | *P* value | aOR (95% CI) | *P* value |
|---|---|---|---|---|---|
| **Dietary supplement** | | | | | |
| Treated vs. Untreated depressive symptoms | 2843 | 1.663 (1.354,2.041) | <0.001* | 1.290 (1.038,1.604) | 0.022* |
| Untreated depressive symptoms vs. No depressive symptoms | 30508 | 0.805 (0.704,0.920) | 0.002* | 0.762(0.632,0.918) | 0.005* |
| Treated depressive symptoms vs. No depressive symptoms | 29485 | 1.338 (1.121,1.596) | 0.001* | 0.850 (0.678,1.065) | 0.155 |
| **Complementary and alternative medicine** | | | | | |
| Treated vs. Untreated depressive symptoms | 2843 | 0.959 (0.772,1.191) | 0.700 | 0.917(0.731,1.150) | 0.450 |
| Untreated depressive symptoms vs. No depressive symptoms | 30508 | 0.589 (0.523,0.663) | <0.001* | 0.869(0.728,1.037) | 0.118 |
| Treated depressive symptoms vs. No depressive symptoms | 29485 | 0.565 (0.468,0.682) | <0.001* | 0.763 (0.598,0.973) | 0.030* |

Note. aOR = adjusted odds ratio. Covariates in the adjusted model include age, gender, diabetes, hypertension, chronic kidney disease, congestive heart failure, and liver disease. Statistical significance of $p < 0.05$ is indicated by *.

## 3.4. Vitamins

There were no significant changes in vitamin usage between those who reported depressive symptoms compared to those who did not report depressive symptoms, or in terms of increasing depressive symptoms severity (S1 Table). Individuals with depressive symptoms had statistically significantly lower odds of taking vitamin B Family or water soluble vitamins (aOR = 0.794, 95% CI: 0.662, 0.953, p = 0.0141; aOR = 0.790, 95% CI: 0.661, 0.943, p = 0.010) and the odds of taking any vitamin class were decreased for each one point increase in PHQ-9 score and was statistically significant (S2 Table). Between those treated and untreated for depressive symptoms, the results were not statistically significant for any individual vitamin but was significant for the fat-soluble vitamin class, where those who were treated for depressive symptoms had 1.339 odds of taking a fat-soluble vitamin (Table 3 and S4 Table). For those with untreated depressive symptoms, compared to those with no depressive symptoms, the results were not statistically significant for any individual vitamins, but use of a vitamin B class supplement was statistically lower (aOR = 0.774, 95% CI: 0.631,0.950, p = 0.015) (Table 3 and S4 Table). Comparing those with treated depressive symptoms to no depressive symptoms, no statistically significant differences were seen in individual vitamins or vitamin classes (Table 3 and S4 Table).

## 4. Discussion

Several randomized clinical trials have indicated the efficacy of dietary supplement inducing vitamin D, B12, magnesium, zinc for reducing depression symptoms [19–24]. Vitamin B family and omega 3, which are precursors to neurotransmitters, are among the most common nutritional deficiencies in individuals with mood disorders [25]. There is evidence that high fish oil consumption can result in low incidence of mental disorders [26]. Our results suggested that individuals with depressive symptoms have lower odds of supplement consumption compared with individuals without depressive symptoms. Taking dietary supplements can indicate various health-related behaviors, as individuals who use supplements are generally less inclined to consume excessive alcohol, smoke, or be overweight [27]. Additionally, they are more prone to participate in physical activities and adopt a more vegetarian eating pattern and to consume more fruits and vegetables [27]. Within the National Diet and Nutrition Survey of British Adults survey, it was observed that individuals who consumed dietary supplements surprisingly displayed higher recorded nutrient intakes solely from food sources compared to those who didn't take supplement [28]. This phenomenon, known as the 'inverse

supplement hypothesis,' suggests that the individuals who are most inclined to use dietary supplements may actually be the ones who require them the least [27, 28]. Previous studies have reported that depression is associated with sedentary lifestyle and lower physical activity [29]. Our findings suggest that supplement users are less likely to have depressive symptoms than non-supplement users. Additionally, it is noteworthy to consider that participants with chronic diseases in this study exhibited higher levels of depressive symptoms. The use of CAM and dietary supplements among these individuals may introduce interactions with their chronic disease medications, which could contribute to the observed depressive symptomatology. Future research should delve deeper into these potential interactions to better understand their impact on mental health outcomes.

Our findings indicate that participants with depressive symptoms who use antidepressants have higher odds of dietary supplement compared with participants with depressive symptoms who do not use antidepressants. This may be due to the potential to self-treat with a range of dietary supplements. Previous studies have demonstrated the possibility of self-medication with over-the-counter (OTC) medicines, CAM and dietary supplements [30, 31]. Additionally, the use of herbal remedies and OTC for self-medication is prevalent among individuals suffering from anxiety and depression [32]. This further implies that the general population may have some level of dissatisfaction with their Western medical treatment, albeit indirectly [30]. This is also in line with prior work that around thirty percent of individuals older than 65 years old may utilize OTC medications alongside their prescribed medicines [30]. We also hypothesize that the higher odds of supplement use in individuals with treated depressive symptoms compared to individuals with untreated depressive symptoms may be due to antidepressants adverse events. An increasing number of studies have shown that antidepressants can induce nutritional depletion and deficiency [33–35]. Hence, individuals might consider utilizing dietary supplements to address the potential effects of antidepressants on their vitamin levels. Finally, the lower odds of dietary supplements use in participants with depressive symptoms who do not use antidepressants may be due their lack of insight or less compliance to antidepressants therapy. Moreover, various factors are involved in supplements and CAM use including reported effectiveness, accessibility, decreased adverse effects, dissatisfaction with current medical treatment, income level, education level, culture. There may also be a selection bias towards using supplements and CAM in people with less severe depression, as people with more severe depression may rely on more biological/interventional treatments such as medication and electroconvulsive therapy [36].

Our study's findings suggested a notable difference in CAM usage between individuals with treated depressive symptoms and those without. This indicated that individuals with treated depressive symptoms may be less inclined to turn to CAM therapies compared to individuals without depressive symptoms. One possibility is that individuals receiving antidepressant treatment for depressive symptoms, may feel that these conventional methods are sufficient for managing their symptoms. They might prioritize the guidance of healthcare professionals and evidence-based antidepressants over CAM approaches. Additionally, the stigma often associated with mental health conditions such as depressive symptoms could play a role. Some individuals might be more comfortable discussing and seeking treatment for their depressive symptoms within the context of traditional healthcare settings, rather than exploring CAM options. Further research should explore the reason for this phenomenon.

It is essential to take into consideration the strengths and limitations inherent in this study. The understanding of the measurement error structure and validity of the NHANES dietary supplements collection methods is currently limited. Conducting validation studies would be necessary to comprehend the extent and types of errors, such as random and systematic errors. Potential sources of measurement error include several factors: 1) participants' recall errors

when reporting all dietary supplement consumed, typical dosage, and frequency within the past 30 days; 2) inaccuracies when interviewers transcribe information from the product label during data collection; 3) misidentification of the reported product with the actual formulation used by the participant; 4) errors during the entry of product label information into the database; and 5) discrepancies between the labeled ingredient amounts and the actual content consumed. Future research endeavors may involve assessing the NHANES dietary supplement collection methods as well as alternative approaches for gathering information on dietary supplement use in surveys and studies. In addition, regarding the definition of depressive symptoms, our study employed the PHQ-9 as a screening tool. While deemed acceptable for screening, it's crucial to acknowledge that the PHQ-9 serves as a screening tool rather than a standardized diagnostic tool. Consequently, individuals identified as MDD through screening, who remain untreated, may not necessarily require antidepressants, potentially not reflecting true MDD cases. Furthermore, the tool might not distinctly capture the level of depressive symptoms, representing a limitation in our study.

The strength of our study is that NHANES is a survey that encompasses a representative sample of the noninstitutionalized population in the US, which adds to its credibility, however it is important to note a limitation. The data utilized in our analysis only extends up to the year 2018 due to the unique circumstances presented by the COVID-19 pandemic and its impact on mental health. This time frame limitation may impact the generalizability of our findings. Finally, it is not possible to completely dismiss the potential influence of self-selection bias, whereby individuals who are more health-conscious may have exhibited greater interest in participating in NHANES [37].

In conclusion, our results indicated significantly lower odds of dietary supplement use in individuals with depressive symptoms compared to adults not reporting depressive symptoms. Notably, individuals receiving treatment for depression demonstrated increased usage of dietary supplements, particularly fat-soluble vitamins such as vitamin D compared with those with depressive symptoms who were not taking antidepressants. Conversely, those with untreated depressive symptoms had lower odds of CAM use compared with participants without depressive symptoms. These findings suggest that treatment may influence supplement intake, especially for essential vitamins. Further replication and validation is needed along with definitive studies to further evaluate the underlying reasons for this phenomenon among individuals with depression. Healthcare providers should engage in discussions regarding the potential benefits of dietary supplements and CAM with patients experiencing depressive symptoms, particularly those not undergoing treatment. Clinicians should consider the impact of treatment on vitamin intake, specifically fat-soluble vitamins, and potentially monitor or recommend appropriate supplementation for individuals with depressive symptoms to ensure adequate nutritional support.

## Supporting information

**S1 Table. Adjusted coefficient estimates (aCoef. Estm) for associations between individual vitamin use and depressive symptoms risk or depressive symptoms severity.**
(DOCX)

**S2 Table. Adjusted ORs for associations between various vitamin classes and depressive symptoms risk or depressive symptoms severity.**
(DOCX)

**S3 Table. Adjusted Coef. Estm for associations between individual vitamin use and depressive symptoms treatments.**
(DOCX)

**S4 Table. Adjusted ORs for associations between various vitamin classes and alternative medicine use and depressive symptoms treatments.**
(DOCX)

## Author Contributions

**Conceptualization:** Shakila Meshkat, Venkat Bhat.

**Data curation:** Michelle Wu.

**Formal analysis:** Michelle Wu.

**Investigation:** Hilary Pang.

**Methodology:** Michelle Wu.

**Supervision:** Hyejung Jung, Wendy Lou, Venkat Bhat.

**Validation:** Michelle Wu.

**Visualization:** Shakila Meshkat, Vanessa K. Tassone.

**Writing – original draft:** Shakila Meshkat, Vanessa K. Tassone.

**Writing – review & editing:** Shakila Meshkat, Vanessa K. Tassone, Hilary Pang, Michelle Wu, Hyejung Jung, Wendy Lou, Venkat Bhat.

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
