## [Decision Letter · Decision Letter 0]

5 Dec 2023

PONE-D-23-21189Adults with Depression Have Lower Odds of Dietary Supplement UsePLOS ONE

Dear Dr. Bhat,

Thank you for submitting your manuscript to PLOS ONE. After careful consideration, we feel that it has merit but does not fully meet PLOS ONE’s publication criteria as it currently stands. Therefore, we invite you to submit a revised version of the manuscript that addresses the points raised during the review process.

We look forward to receiving your revised manuscript.

Kind regards,

Jenny Wilkinson, PhD

Academic Editor

PLOS ONE

Journal Requirements:

"I have read the journal's policy and the authors of this manuscript have the following

competing interests: VB is supported by an Academic Scholar Award from the

University of Toronto Department of Psychiatry and has received research support

from the Canadian Institutes of Health Research, Brain & Behavior Foundation, Ontario

Ministry of Health Innovation Funds, Royal College of Physicians and Surgeons of

Canada, Department of National Defence (Government of Canada), New Frontiers in

Research Fund, Associated Medical Services Inc. Healthcare, American Foundation

for Suicide Prevention, Roche Canada, Novartis, and Eisai."

Additional Editor Comments:

Thank you for your submission. Reviews are now complete and reviewers comments are provided. Both reviewers have highlighted significant areas for review in the work particularly around the use, analysis and interpretation of the data and have provided suggestions for improvements. you are invited to consider these comments and provide a significant revision of your work to address the concerns raised by the reviewers.

Reviewers' comments:

Reviewer's Responses to Questions

**Comments to the Author**

1. Is the manuscript technically sound, and do the data support the conclusions?

Reviewer #1: No

Reviewer #2: No

2. Has the statistical analysis been performed appropriately and rigorously? 

Reviewer #1: Yes

Reviewer #2: No

3. Have the authors made all data underlying the findings in their manuscript fully available?

Reviewer #1: Yes

Reviewer #2: Yes

4. Is the manuscript presented in an intelligible fashion and written in standard English?

Reviewer #1: Yes

Reviewer #2: Yes

5. Review Comments to the Author

Reviewer #1: This manuscript utilized data from NHANES 2005-2018 to address the question whether adults with depression use nutritional supplements to a different degree than non-depressed adults. This is an interesting question as there is evidence that supplement use can be used successfully as adjunct therapy in depression. Using this large, representative sample is a strength of the study; however, researchers are limited to the data collected to direct their question. My main concern with this work is how the outcome variables were defined (part 2.3). The investigators defined dietary supplement use by reporting individual nutrient use: B1, B2, B3, B6, B12, vitamins C, K, and D, folic acid, folate choline, calcium, P, Mg, Fe, Zn, Cu, Na, K, Se, and caffeine. Additionally, nutrients were grouped as water soluble vitamins, fat soluble vitamins, and B vitamins. CAM was defined as folate, Mg, B6, B9, D, Zn, and Se. Hence, there is tremendous overlap in the data presented, and the results, thus, are convoluted and repetitive. Also, it is unclear if multivitamin/mineral use is included in all of these categories, or if only individual supplementation is examined. A more systematic approach (with no overlap) would produce more useful, meaningful results. Often multivitamin/mineral use is separated from single nutrient supplementation. In addition to a multivitamin/mineral supplement, the top 3-5 supplements could be investigated (with an ‘other’ group). For fat soluble vitamins, only D and K are examined – yet E is one of the leading individual nutrients supplemented in the US. The omega 3 fats are also at the top of the list of supplemented nutrients and are not examined in this report (although mentioned in the discussion). Why is caffeine in this analysis? Furthermore, there is no data presented on caffeine as well as several other nutrients listed above. If variables are stated as outcomes in the methods section, they need to be reported in the results section.

In the text, CAM is defined (lines 79-86) as a range of medical and healthcare systems, practices, and products – including herbal agents, and exercise. Yet, in methods CAM is defined as folate, Mg, B6, B9 (which is folate – thus repetitive), D, Zn, and Se. There is no justification for this definition of CAM and it is not consistent with the accepted definition of CAM. See: https://www.cancer.gov/about-cancer/treatment/cam Also, the authors discuss the use of OTC – yet they did not look at these data in the NHANES data set. The authors need to be more cautious in their approach to the research question.

Since herbal agents, exercise, and omega 3 fats are in the NHANES data set, it is unclear why these variables were not assessed in this investigation.

Other concerns:

• The introduction is sourced mainly from reviews and websites. There is only a cursory overview of the topic with little primary research cited. Depending on second-hand sources to such a large degree lowers the confidence of the reader for accepting the rational provided to justify the investigation. Include primary data in the introduction to support statements.

• Line 71: what is meant by ‘regular’ diet?

• Section 2.1: provide citations for the characteristics of NHANES data.

• Line 105: ‘need to sign’ to replace ‘have to sign’

• Line 123: what is “249” – there is no definition or citation.

• Throughout the text, do not capitalize nutrients – these are not proper nouns.

• Line 145: provide a citation

• All tables should have a legend to explain the presentation of the data and the p value.

• Table 1: For ‘sex’ add a line for male data (similar to how there is a line for each component of ‘race’).

• Table 1: the final 6 lines need to be reconsidered as discussed above. (similar comment for tables 4-6)

• Line 213: replace ‘levels’ with ‘usage’

• Discussion: the first paragraph can be deleted as it is a repeat of the results.

• Line 316: need to add ‘compared to adults not reporting depression’. The comparison group needs to be identified.

Reviewer #2: Thank you for the opportunity to review this paper

1. The authors referenced line 81 to justify CAM's indication for depression, citing source number 14. However, upon further examination of this source, it does not distinctly indicate the role of CAM in treating depression. Referencing the article by Ng, J.Y., Nazir, Z. & Nault, H. (2020) on "Complementary and alternative medicine recommendations for depression: a systematic review and assessment of clinical practice guidelines," it was observed that the quality of CAM recommendations was generally lower. Therefore, it is crucial to narrow down the scope of CAM that has proven to be beneficial for depression.

2. In reference to line 99 mentioning the use of NHANES data covering participants from 20 to 80 years old, a 2020 US report highlights varying prevalence rates of depression across age groups, with the highest among 18–24 years (21.5%) and the lowest among those ≥65 years (14.2%). However, the data analyzed in this study covers adults from 18 to 80 years old (line 166). Clarification is needed regarding the real age range within this dataset. Additionally, it's important to note that this dataset only covers information until 2018, which presents a limitation. The reason for not utilizing more current NHANES data should be addressed.

3. Concerning the definition of MDD, the use of PHQ-9 (with a sensitivity and specificity cutoff of 10 at more than 85%) is deemed acceptable as a screening tool rather than a standardized diagnostic tool. However, it should be acknowledged that individuals screened as MDD and remaining untreated might not necessarily require antidepressants, potentially not reflecting true MDD cases. Furthermore, the tool might not distinctly capture the level of depression, which should be highlighted as a limitation.

4. Further examination of the CAM definition reveals that it encompasses more than just dietary supplements, contradicting the initial understanding. Notably, at line 133, B6, vitamin D, and Zinc are also considered part of CAM. To maintain consistency throughout the manuscript, a single term, preferably "dietary supplement," should be used instead of alternating between CAM and dietary supplement. Please refer to reference at question 1 for the various definition of CAM

5. Certain variables within the NHANES dataset might act as confounders. For instance, the presence of chronic diseases like diabetes could influence the decision to take dietary supplements rather than using them specifically for treating depression. It's essential to explore whether the authors investigated other potential covariates or confounders that are available in the dataset.

6. The method for defining depression severity is unclear. Specifically, how did the authors delineate the severity using the raw total score of PHQ-9? What cutoffs were employed to categorize mild, moderate, and severe depression? Line 155 mentions analyzing depression severity using logistic regression. Clarification is needed on the cutoffs and grouping used for the analysis of dietary supplement and depression severity.

7. Due to the unclear definition of certain variables, it would be inappropriate to draw conclusions solely based on the results.

6. PLOS authors have the option to publish the peer review history of their article (what does this mean?). If published, this will include your full peer review and any attached files.

Reviewer #1: No

Reviewer #2: No

---

## [Author Response · Author response to Decision Letter 0]

22 Jan 2024

Response to reviewers

Reviewer #1: This manuscript utilized data from NHANES 2005-2018 to address the question whether adults with depression use nutritional supplements to a different degree than non-depressed adults. This is an interesting question as there is evidence that supplement use can be used successfully as adjunct therapy in depression. Using this large, representative sample is a strength of the study; however, researchers are limited to the data collected to direct their question. 

My main concern with this work is how the outcome variables were defined (part 2.3). The investigators defined dietary supplement use by reporting individual nutrient use: B1, B2, B3, B6, B12, vitamins C, K, and D, folic acid, folate choline, calcium, P, Mg, Fe, Zn, Cu, Na, K, Se, and caffeine. Additionally, nutrients were grouped as water soluble vitamins, fat soluble vitamins, and B vitamins. CAM was defined as folate, Mg, B6, B9, D, Zn, and Se. Hence, there is tremendous overlap in the data presented, and the results, thus, are convoluted and repetitive. Also, it is unclear if multivitamin/mineral use is included in all of these categories, or if only individual supplementation is examined. 

A more systematic approach (with no overlap) would produce more useful, meaningful results. Often multivitamin/mineral use is separated from single nutrient supplementation. In addition to a multivitamin/mineral supplement, the top 3-5 supplements could be investigated (with an ‘other’ group). For fat soluble vitamins, only D and K are examined – yet E is one of the leading individual nutrients supplemented in the US. The omega 3 fats are also at the top of the list of supplemented nutrients and are not examined in this report (although mentioned in the discussion). 

Why is caffeine in this analysis? Furthermore, there is no data presented on caffeine as well as several other nutrients listed above. If variables are stated as outcomes in the methods section, they need to be reported in the results section.

Response: Thank you for your valuable feedback. We appreciate the insightful comments provided by Reviewer #1. In response to your concerns, we have made several revisions to enhance the clarity and relevance of our study.

1. Multivitamin/Mineral Use: We have incorporated 'multivitamin/mineral use' into our dietary supplement analysis, providing a more comprehensive perspective on supplement intake.

2. Inclusion of Missing Nutrients: Based on your suggestion, we have expanded our analysis to include Vitamin E and Omega-3 Fatty Acids.

3. Exclusion of Caffeine: We acknowledge the oversight and have removed caffeine from the analysis.

4. Refinement of CAM Definition: To address the issue of repetitiveness, we have redefined CAM based on the available NHANES data. We acknowledge the absence of other CAM agents including herbal agents data and, unfortunately, NHANES does not provide this information. The revised definition of CAM now includes: ‘’Selected CAM agents were: folate, magnesium, vitamin B6, B9, D, zinc, selenium, and omega-3 fatty acids. In this study, CAM was defined as a binary variable of “Yes” if any of the above CAM agents were taken or if an individual participated in >150 minutes of physical activity a week, and “No” otherwise.’’

In the text, CAM is defined (lines 79-86) as a range of medical and healthcare systems, practices, and products – including herbal agents, and exercise. Yet, in methods CAM is defined as folate, Mg, B6, B9 (which is folate – thus repetitive), D, Zn, and Se. There is no justification for this definition of CAM and it is not consistent with the accepted definition of CAM. See: https://www.cancer.gov/about-cancer/treatment/cam Also, the authors discuss the use of OTC – yet they did not look at these data in the NHANES data set. The authors need to be more cautious in their approach to the research question.

Since herbal agents, exercise, and omega 3 fats are in the NHANES data set, it is unclear why these variables were not assessed in this investigation.

Response: Thank you for bringing these concerns to our attention. We have redefined CAM based on the available NHANES data. We acknowledge the absence of other CAM agents including herbal agents data and, unfortunately, NHANES does not provide this information. The revised definition of CAM now includes: ‘’Selected CAM agents were: folate, magnesium, vitamin B6, B9, D, zinc, selenium, and omega-3 fatty acids. In this study, CAM was defined as a binary variable of “Yes” if any of the above CAM agents were taken or if an individual participated in >150 minutes of physical activity a week, and “No” otherwise.’’

Thank you for providing clarification on the dataset limitations related to OTC medications. We want to acknowledge your comment and clarify that the NHANES dataset specifically includes information on prescribed medications only, and unfortunately, OTC data is not available. 

Other concerns:

• The introduction is sourced mainly from reviews and websites. There is only a cursory overview of the topic with little primary research cited. Depending on second-hand sources to such a large degree lowers the confidence of the reader for accepting the rational provided to justify the investigation. Include primary data in the introduction to support statements.

Response: Thank you for your comment. We have incorporated the original studies as references in the introduction section.

• Line 71: what is meant by ‘regular’ diet?

Response: Thank you for your comment. In the context of our statement, a "regular" diet refers to a balanced and nutritionally adequate diet that provides essential nutrients in appropriate proportions. We revised that section of the manuscript.

‘’Despite the fact that dietary supplements are not recommended for individuals who are healthy and consume a well-balanced and nutritionally adequate diet, the industry generates an annual revenue of 1.2 billion USD.’’

• Section 2.1: provide citations for the characteristics of NHANES data.

Response: Thank you for your comment. We added a citation for the characteristics of NHANES data. 

‘’Zipf G, Chiappa M, Porter KS, Ostchega Y, Lewis BG, Dostal J. National Health and Nutrition Examination Survey: Plan and Operations, 1999–2010. Vital Health Stat. 2013;1(56):37.’’

• Line 105: ‘need to sign’ to replace ‘have to sign’

Response: We appreciate your keen observation. In response to your comment, we have made the suggested change, replacing 'have to sign' with 'need to sign'. 

• Line 123: what is “249” – there is no definition or citation.

Response: Thank you for your comment. 249 is how NHANES coded antidepressants in the Prescription Medications - Drug Information Questionnaire questionnaire. We added ‘’https://wwwn.cdc.gov/Nchs/Nhanes/1999-2000/RXQ_DRUG.htm’’ to the methods section for further clarification.

• Throughout the text, do not capitalize nutrients – these are not proper nouns.

Response: Thank you for your comment. In response to your feedback, we have made the necessary corrections in the revised manuscript to ensure that nutrients are not capitalized

• Line 145: provide a citation

Response: Thank you for your comment. We provide citations for line 145 in the revised manuscript. 

• All tables should have a legend to explain the presentation of the data and the p value.

Response: Thank you for your comment. We added legends to all tables to further explain the data. 

• Table 1: For ‘sex’ add a line for male data (similar to how there is a line for each component of ‘race’).

Respons: Thank you for your comment. In response to your comment, we have added a dedicated line in Table 1, providing a more comprehensive presentation of both male and female participant numbers.

• Table 1: the final 6 lines need to be reconsidered as discussed above. (similar comment for tables 4-6)

Response: Thank you for your comment. We redefined these variables in the revised manuscript. 

• Line 213: replace ‘levels’ with ‘usage’

Response: Thank you for your comment. In response to your comment, we have made the suggested change, replacing 'levels' with '‘usage’'. 

• Discussion: the first paragraph can be deleted as it is a repeat of the results.

Response: Thank you for your comment. We deleted the first paragraph of discussion in the revised manuscript.

• Line 316: need to add ‘compared to adults not reporting depression’. The comparison group needs to be identified.

Response: Thank you for your comment. In response to your comment, we have revised the sentence on line 316 to explicitly state, ‘’In conclusion, our results indicated significantly lower odds of dietary supplement use in individuals with depressive symptoms compared to adults not reporting depressive symptoms.’’

\fReviewer #2: Thank you for the opportunity to review this paper

1. The authors referenced line 81 to justify CAM's indication for depression, citing source number 14. However, upon further examination of this source, it does not distinctly indicate the role of CAM in treating depression. Referencing the article by Ng, J.Y., Nazir, Z. & Nault, H. (2020) on "Complementary and alternative medicine recommendations for depression: a systematic review and assessment of clinical practice guidelines," it was observed that the quality of CAM recommendations was generally lower. Therefore, it is crucial to narrow down the scope of CAM that has proven to be beneficial for depression.

Response: We appropriate the reviewer for their insightful comment. We revised that section based on your feedback: ‘’Despite the advantages associated with CAM, such as a lower incidence of adverse events, reported efficacy, a comprehensive perspective on individual problems, and dissatisfaction with traditional care, the quality of CAM recommendations for depression has been noted to be generally lower.’’

2. In reference to line 99 mentioning the use of NHANES data covering participants from 20 to 80 years old, a 2020 US report highlights varying prevalence rates of depression across age groups, with the highest among 18–24 years (21.5%) and the lowest among those ≥65 years (14.2%). However, the data analyzed in this study covers adults from 18 to 80 years old (line 166). Clarification is needed regarding the real age range within this dataset. Additionally, it's important to note that this dataset only covers information until 2018, which presents a limitation. The reason for not utilizing more current NHANES data should be addressed.

Response: Thank you for your insightful comment. We have addressed the concern about the age range in the revised manuscript, providing clarification on the NHANES data source. The corrected statement now reads:

‘’NHANES is a survey based on nationally representative data from civilian, non-institutionalized populations aged 18-80 years across 50 states of the United States, utilizing a complex, multistage, probability sampling design.’’ 

Furthermore, we acknowledge the limitation associated with the dataset's timeframe, as it covers information up to 2018. In the revised manuscript, we have explicitly mentioned the reason for not utilizing more recent NHANES data. Due to the unprecedented impact of the COVID-19 pandemic on mental health and depression, we made the decision to focus on data up to 2018 to maintain consistency and reliability in our analysis. 

‘’The strength of our study is that NHANES is a survey that encompasses a representative sample of the noninstitutionalized population in the US, which adds to its credibility, it is important to note a limitation. The data utilized in our analysis only extends up to the year 2018 due to the unique circumstances presented by the COVID-19 pandemic and its impact on mental health. This time frame limitation may impact the generalizability of our findings.’’

3. Concerning the definition of MDD, the use of PHQ-9 (with a sensitivity and specificity cutoff of 10 at more than 85%) is deemed acceptable as a screening tool rather than a standardized diagnostic tool. However, it should be acknowledged that individuals screened as MDD and remaining untreated might not necessarily require antidepressants, potentially not reflecting true MDD cases. Furthermore, the tool might not distinctly capture the level of depression, which should be highlighted as a limitation.

Response: Thank you for your insightful comment. We appreciate the importance of clarifying the nature of the PHQ-9 as a screening tool. To accurately reflect this distinction, we have modified instances of 'depression' to 'depressive symptoms' throughout the manuscript, including the title. Additionally, we have incorporated this clarification into the limitation section of the manuscript to transparently communicate the nature of our assessment tool and its implications for the study's findings: 

‘’In addition, regarding the definition of depressive symptoms, our study employed the PHQ-9 as a screening tool, utilizing a sensitivity and specificity cutoff of 10 at more than 85%. While deemed acceptable for screening, it's crucial to acknowledge that the PHQ-9 serves as a screening tool rather than a standardized diagnostic tool. Consequently, individuals identified as MDD through screening, who remain untreated, may not necessarily require antidepressants, potentially not reflecting true MDD cases. Furthermore, the tool might not distinctly capture the level of depressive symptoms, representing a limitation in our study.’’

4. Further examination of the CAM definition reveals that it encompasses more than just dietary supplements, contradicting the initial understanding. Notably, at line 133, B6, vitamin D, and Zinc are also considered part of CAM. To maintain consistency throughout the manuscript, a single term, preferably "dietary supplement," should be used instead of alternating between CAM and dietary supplement. Please refer to reference at question 1 for the various definition of CAM.

Response: Thank you for your comment. We have redefined CAM based on the available NHANES data. We acknowledge the absence of other CAM agents including herbal agents data and, unfortunately, NHANES does not provide this information. The revised definition of CAM now includes: ‘’Selected CAM agents were: folate, magnesium, vitamin B6, B9, D, zinc, selenium, and omega-3 fatty acids. In this study, CAM was defined as a binary variable of “Yes” if any of the above CAM agents were taken or if an individual participated in >150 minutes of physical activity a week, and “No” otherwise.’’

5. Certain variables within the NHANES dataset might act as confounders. For instance, the presence of chronic diseases like diabetes could influence the decision to take dietary supplements rather than using them specifically for treating depression. It's essential to explore whether the authors investigated other potential covariates or confounders that are available in the dataset.

Response: Thank you for your comment. In response, we have expanded our analysis by incorporating additional covariates (chronic diseases). The updated set of covariates now includes age, gender, diabetes, hypertension, chronic kidney disease, congestive heart failure, and liver disease.

6. The method for defining depression severity is unclear. Specifically, how did the authors delineate the severity using the raw total score of PHQ-9? What cutoffs were employed to categorize mild, moderate, and severe depression? Line 155 mentions analyzing depression severity using logistic regression. Clarification is needed on the cutoffs and grouping used for the analysis of dietary supplement and depression severity.

Response: Thank you for your comment. We did not categorize depressive symptoms severity into mild/moderate/severe due to small sample sizes. We added ‘’Depressive symptoms severity was defined as just the raw PHQ-9 score, where a higher score indicates more depressive symptoms.’’ to further clarity how we defined depressive symptoms severity.

7. Due to the unclear definition of certain variables, it would be inappropriate to draw conclusions solely based on the results.

Response: Thank you for your insightful comment. In response, we have taken steps to enhance the clarity of our study by redefining certai

---

## [Decision Letter · Decision Letter 1]

15 Feb 2024

PONE-D-23-21189R1Adults with Depressive Symptoms Have Lower Odds of Dietary Supplement UsePLOS ONE

Dear Dr. Bhat,

Thank you for submitting your manuscript to PLOS ONE. After careful consideration, we feel that it has merit but does not fully meet PLOS ONE’s publication criteria as it currently stands. Therefore, we invite you to submit a revised version of the manuscript that addresses the points raised during the review process. The comments of the reviewers are attached for your information, these comments are seeking clarity on some aspects of your study and provided suggestions from strengthening the work.

We look forward to receiving your revised manuscript.

Kind regards,

Jenny Wilkinson, PhD

Academic Editor

PLOS ONE

Journal Requirements:

**Additional Editor Comments:**

Thank you for your responses and revisions to the manuscript. Once reviewer has provided some further comments in relation to the revisions and you are invited to provide a response to these comments.

Reviewers' comments:

Reviewer's Responses to Questions

**Comments to the Author**

1. If the authors have adequately addressed your comments raised in a previous round of review and you feel that this manuscript is now acceptable for publication, you may indicate that here to bypass the “Comments to the Author” section, enter your conflict of interest statement in the “Confidential to Editor” section, and submit your "Accept" recommendation.

Reviewer #1: All comments have been addressed

Reviewer #2: (No Response)

2. Is the manuscript technically sound, and do the data support the conclusions?

Reviewer #1: Yes

Reviewer #2: Partly

3. Has the statistical analysis been performed appropriately and rigorously? 

Reviewer #1: Yes

Reviewer #2: No

4. Have the authors made all data underlying the findings in their manuscript fully available?

Reviewer #1: Yes

Reviewer #2: Yes

5. Is the manuscript presented in an intelligible fashion and written in standard English?

Reviewer #1: Yes

Reviewer #2: Yes

6. Review Comments to the Author

Reviewer #1: (No Response)

Reviewer #2: Thank you for the opportunity to review this revision. I have some concern regarding methodology, result and discussion

1. It is still essential to use the recent data. If the authors are concern about the COVID, it can be adjusted with statistical method treating COVID as confounding or making a subgroup analysis of this nested data for more relevant and novel result

2. Table 1 one elaborates on the characteristic of the participants. It is obvious that those people with chronic disease tend to have depressive symptom, However it was not briefly discussed in the result and discussion section. Just add some statement that using CAM and dietary supplement may also have interaction with the chronic disease medication

3. It It will be more meaningful to classify the severity of disease rather than using the raw score of the PHQ9 so the aOR would be more meaningful to translate. As a reader will prefer to see the aOR of severity level rather than aOR for each 1 unit of PHQ9. Table 2,4 and 5 on depressive severity have aOR of around 0.9 which is quite smaller. To accommodate this, logistic regression is preferred. We might see more meaningful association of particular CAM and supplement such as vitamin D

4. Also the authors talked about depressive symptoms which is treated as binary using cutoff of 10 (<10 is no and >10 is yes). Why didn't the authors used logistic regression for table 4 and 6 and just present the odd ratio of table 5 and 7 to make the manuscript more concise without abundant coefficients tables?

5. In characteristic table 1, age was not different between group, but race was different. Why the adjusted model did not use race but put age instead?

6. Is it still fall into study objective if the authors compare the treated individual vs non depressive individual, which is missing in the manuscript?

7. Again, this manuscript reveals that adult with depressive symptoms have lower use of cam and supplement, although in the introduction, some CAMs are not clinically recommended. Looking at table 6 and 7 the intake of fat soluble vitamin which also cover vitamin D was significantly higher in treated depressive individual Which is good. With these conflicting results, what would be the final conclusion and recommendations?

7. PLOS authors have the option to publish the peer review history of their article (what does this mean?). If published, this will include your full peer review and any attached files.

Reviewer #1: No

Reviewer #2: No

---

## [Author Response · Author response to Decision Letter 1]

3 Apr 2024

Response to reviewers

1. It is still essential to use the recent data. If the authors are concern about the COVID, it can be adjusted with statistical method treating COVID as confounding or making a subgroup analysis of this nested data for more relevant and novel result.

Response: Thank you for your valuable feedback. We appreciate your suggestion regarding adjusting for COVID effects using statistical methods. However, the data is not avaible (please see: 2019-2020:not representative- https://wwwn.cdc.gov/nchs/nhanes/continuousnhanes/default.aspx?BeginYear=201; 2017-March 2020 Pre-pandemic: Not completed- https://wwwn.cdc.gov/nchs/nhanes/continuousnhanes/default.aspx?Cycle=2017-2020;" https://wwwn.cdc.gov/nchs/nhanes/continuousnhanes/default.aspx?Cycle=2017-2020; 2021-2023:Not available- https://wwwn.cdc.gov/nchs/nhanes/continuousnhanes/default.aspx?Cycle=2017-2020). Furthermore, we believe that adjusting for all potential COVID effects might not be feasible due to several reasons. Firstly, the NHANES dataset does not include specific questions regarding the history of COVID-19 infection or treatments received. Secondly, COVID-19 has had multifaceted impacts on health outcomes, including indirect effects on lifestyle, mental health, and healthcare access, which are not easily quantifiable within the NHANES dataset. Additionally, using data after the COVID-19 period might introduce further challenges. Post-COVID data may be confounded by temporal changes in lifestyle and health-seeking behavior, limited availability of such data depending on the timing of the study, continued uncertainty and variability of pandemic effects, potential shifts in dietary patterns during the pandemic, the profound impact of the pandemic on mental health, and ethical considerations surrounding research conducted immediately after a major global health crisis. 

2. Table 1 one elaborates on the characteristic of the participants. It is obvious that those people with chronic disease tend to have depressive symptom, However it was not briefly discussed in the result and discussion section. Just add some statement that using CAM and dietary supplement may also have interaction with the chronic disease medication.

Response: Thank you for your comment. We added ‘’Additionally, It is noteworthy to consider that participants with chronic diseases in this study exhibited higher levels of depressive symptoms. The use of CAM and dietary supplements among these individuals may introduce interactions with their chronic disease medications, which could contribute to the observed depressive symptomatology. Future research should delve deeper into these potential interactions to better understand their impact on mental health outcomes.’’ to the discussion section.

3. It It will be more meaningful to classify the severity of disease rather than using the raw score of the PHQ9 so the aOR would be more meaningful to translate. As a reader will prefer to see the aOR of severity level rather than aOR for each 1 unit of PHQ9. Table 2,4 and 5 on depressive severity have aOR of around 0.9 which is quite smaller. To accommodate this, logistic regression is preferred. We might see more meaningful association of particular CAM and supplement such as vitamin D

Response: Thank you for your comment. We first categorized depressive symptoms into 'Yes (at risk)' and 'No,' in hopes of simplifying the understanding of the risk of depressive symptoms and CAM intake. We agree that additional insight can be drawn with severity of disease though it was not possible to run such models reliably due to the small number of individuals in some of our categories. Instead we had included the PHQ9 as a continuous score to allow us to observe trends without losing statistical power that may have happened when dichotomizing our depressive symptom scores. 

4. Also the authors talked about depressive symptoms which is treated as binary using cutoff of 10 (<10 is no and >10 is yes). Why didn't the authors used logistic regression for table 4 and 6 and just present the odd ratio of table 5 and 7 to make the manuscript more concise without abundant coefficients tables?

Response: Thank you for your comment. We would like to clarify that we did not use logistic regression for Table 4 and 6 as the outcome was specific vitamin intake (ex. mg, mcg) which were treated as continuous variables. Linear regressions were used for each vitamin intake where depressive symptoms were included as one of the covariates. Based on your comment, we moved tables 4-7 to supplementary materials to avoid having abundant tables. 

5. In characteristic table 1, age was not different between group, but race was different. Why the adjusted model did not use race but put age instead?

Response: Thank you for your comment. We added ethnicity to covariates and re-analyzed the data. 

6. Is it still fall into study objective if the authors compare the treated individual vs non depressive individual, which is missing in the manuscript?

Response: Thank you for your comment. We compare the treated individual vs non depressive individual analysis results to the revised manuscript.

‘’No statistically significant association was seen between those with treated depressive symptoms versus no depressive symptoms in terms of dietary supplement usage (Table 3).’’

‘’In those with treated depressive symptoms compared to no depressive symptoms, CAM use was lower in comparison (aOR = 0.763, 95% CI = 0.598,0.973, p = 0.030) (Table 3).’’

We also added ‘’Our study's findings suggested a notable difference in CAM usage between individuals with treated depressive symptoms and those without. This indicated that individuals with treated depressive symptoms may be less inclined to turn to CAM therapies compared to individuals without depressive symptoms. One possibility is that individuals receiving antidepressant treatment for depressive symptoms, may feel that these conventional methods are sufficient for managing their symptoms. They might prioritize the guidance of healthcare professionals and evidence-based antidepressants over CAM approaches. Additionally, the stigma often associated with mental health conditions such as depressive symptoms could play a role. Some individuals might be more comfortable discussing and seeking treatment for their depressive symptoms within the context of traditional healthcare settings, rather than exploring CAM options. Further research should explore the reason for this phenomenon.’’ to the discussion section of the revised manuscript.

7. Again, this manuscript reveals that adult with depressive symptoms have lower use of cam and supplement, although in the introduction, some CAMs are not clinically recommended. Looking at table 6 and 7 the intake of fat soluble vitamin which also cover vitamin D was significantly higher in treated depressive individual Which is good. With these conflicting results, what would be the final conclusion and recommendations?

Response: Thank you for your comment. Based on your comment we added ‘’In conclusion, our results indicated significantly lower odds of dietary supplement use in individuals with depressive symptoms compared to adults not reporting depressive symptoms. Notably, individuals receiving treatment for depression demonstrated increased usage of dietary supplements, particularly fat-soluble vitamins such as vitamin D compared with those with depressive symptoms who were not taking antidepressants. Conversely, those with untreated depressive symptoms had lower odds of CAM use compared with participants without depressive symptoms. These findings suggest that treatment may influence supplement intake, especially for essential vitamins. Further replication and validation is needed along with definitive studies to further evaluate the underlying reasons for this phenomenon among individuals with depression. Healthcare providers should engage in discussions regarding the potential benefits of dietary supplements and CAM with patients experiencing depressive symptoms, particularly those not undergoing treatment. Clinicians should consider the impact of treatment on vitamin intake, specifically fat-soluble vitamins, and potentially monitor or recommend appropriate supplementation for individuals with depressive symptoms to ensure adequate nutritional support.’’ to the revised manuscript.

---

## [Editor Report · Decision Letter 2]

10 Apr 2024

Adults with Depressive Symptoms Have Lower Odds of Dietary Supplement Use

PONE-D-23-21189R2

Dear Dr. Bhat,

We’re pleased to inform you that your manuscript has been judged scientifically suitable for publication and will be formally accepted for publication once it meets all outstanding technical requirements.

Kind regards,

Jenny Wilkinson, PhD

Academic Editor

PLOS ONE

Additional Editor Comments (optional):

Thank you for your responses to reviewer's comments. These and the manuscript revisions have satisfactorily addressed the reviewer concerns.